# Bicycle Speed Modelling Considering Cyclist Characteristics, Vehicle Type and Track Attributes

Xingchen Yan [1], Xiaofei Ye [2], Jun Chen [3,4,*], Tao Wang [5], Zhen Yang [1] and Hua Bai [6]

1   College of Automobile and Traffic Engineering, Nanjing Forestry University, Longpan Road 159#, Nanjing 210037, China; xingchenyan.acad@gmail.com (X.Y.); zyang_2016@163.com (Z.Y.)
2   School of Maritime and Transportation, Ningbo University, Fenghua Road 818#, Ningbo 315211, China; yexiaofei@nbu.edu.cn
3   School of Transportation, Southeast University, Dongnandaxue Road 2#, Jiangning Development Zone, Nanjing 211189, China
4   National Demonstration Center for Experimental Road and Traffic Engineering Education (Southeast University), Dongnandaxue Road 2#, Jiangning Development Zone, Nanjing 211189, China
5   School of Architecture and Transportation, Guilin University of Electronic Technology, Jinji Road 1#, Guilin 541004, China; wangtao_seu@163.com
6   China Design Group Co., Ltd., Ziyun Road 9#, Nanjing 210014, China; bh_birch@163.com
*   Correspondence: chenjun@seu.edu.cn; Tel.: +86-139-1394-5222

**Abstract:** Cycling is an increasingly popular mode of transport as part of the response to air pollution, urban congestion, and public health issues. The emergence of bike sharing programs and electric bicycles have also brought about notable changes in cycling characteristics, especially cycling speed. In order to provide a better basis for bicycle-related traffic simulations and theoretical derivations, the study aimed to seek the best distribution for bicycle riding speed considering cyclist characteristics, vehicle type, and track attributes. *K*-means clustering was performed on speed subcategories while selecting the optimal number of clustering using *L* method. Then, 15 common models were fitted to the grouped speed data and Kolmogorov–Smirnov test, Akaike information criterion, and Bayesian information criterion were applied to determine the best-fit distribution. The following results were acquired: (1) bicycle speed sub-clusters generated by the combinations of bicycle type, bicycle lateral position, gender, age, and lane width were grouped into three clusters; (2) Among the common distribution, generalized extreme value, gamma and lognormal were the top three models to fit the three clusters of speed dataset; and (3) integrating stability and overall performance, the generalized extreme value was the best-fit distribution of bicycle speed.

**Keywords:** bicycling characteristics; speed modelling; *K*-means clustering; *L* method; distribution model; model comparison

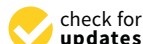

## 1. Introduction

With air pollution, urban congestion, and public health issues like obesity becoming a concern, cycling is an increasingly popular mode of transport as part of the response. The widespread growth of bike-sharing programs and electric bikes across the world invigorate bicycle travel [1]. The emergence of new forms and vehicle types have also brought about many changes in cycling characteristics, which is more significant in the rise of bicycle speed. The increase of vehicle types combining with the traditional impact factors like cyclist and cycling track attributes make bicycle speed more heterogeneous. Therefore, there is a need to renew the appropriate mathematical distributions for cycling speed.

The rapid growth of electric bicycle (EB) transforms the constitution of bicycle flow, from a pure flow consisting of only conventional bicycles (CB) to two types of bikes including EB. This notably increases the heterogeneity of bike riding speed. In a relatively early study by Cherry in 2007 [2], he investigated the speed distributions of CB and EB and analyzed the difference. He found EB ran about 40% faster than CB without speed limit

while 30% with speed limit. In 2008, Lin et al. [3] conducted a specific comparison study on operating speed of CB and EB. Their study presented that EB's speed was 47.6% higher than CB's, close to the result of Cherry. Further, they applied several models to fit bicycles' speed and concluded logarithmic normal distribution was the best-fit distribution. Jin et al. [4] also analyzed the speed difference between CB and EB. Moreover, the gender, age, and loading state's impact on speed were quantitatively compared. Afterwards, Jin et al. [5] performed a speed–flow relationship research for bicycle flow and they estimated the bicycle free flow speed using linear regression method. In 2017, Xu et al. [6] used a Gaussian mixture model to fit the speed distribution of mixed bicycle flow, and they recommended three-component model for free flow and five- or six- component model for other conditions. Xu et al. proposed speed limits for heterogeneous bicycle flow under different widths in their later study [7]. For the influence factors of bicycle speed, besides the bicycle type [2], characteristics of cyclists (gender, age) [3,4], roadway attributes (lane width, grade) [8,9], loading state of vehicles [4] have been studied by researchers. Gender and age were the most common factors. In the previous research, the male and young cyclists with higher speed were observed while the female and old groups were slower. In respect to roadway attributes, bicycle speed presented a positive relationship with lane width while keeping a negatively related with road gradient. For loading state, it impacted bicycle speed in most cases.

Together with the common factors like rider characteristics, facility attributes, et al., the development of EB increases the dispersion and diversity of bicycle speed considerably. The existing studies mainly focuses on the analysis of the speed difference between CB and EB. Lin et al. and Xu et al. tried some unimodal distributions and Gaussian mixture distribution to fit bicycle speed respectively. The former modelled the distribution of speeds only considering bike type and the later established a refined mixture model taking more factors into considerations. However, due to factor bike volume in Xu et al.'s model, the number of subcategories, weighting index, and other factors as well as their relationships to each other need to be determined and further explored. Considering volume factor in speed modelling yields too many cases and compositions of bicycle flow. In this situation, it is hard to capture the natural characteristics of bicycles, which is the basis for traffic simulations (e.g., speed initialization) and theoretical derivations (e.g., setting the input parameters of a model).

In sum, bicycle riding speed is influenced by as many factors as the presence of electric bicycles. These influence factors should be taken into account in modelling processes whereas the difficulty of modelling and the complexity of models will increase meanwhile. To achieve a balance, the present study aims to establish relatively simplified models when considering the most common basic factors impacting bicycle speed. To obtain fewer cases and capture the nature of cycling speed, we only consider the cycling speed in free flow in which the riding state of a bike is influenced by rider personal features, vehicle performance, and road attributes, rather than any other disturbances, and use clustering to reduce the subcategories resulting from the basic factors. Meanwhile, to best fit cycling speed, 15 usual continuous distribution models are tested and compared.

The rest of this paper consists of five sections. Section 2 indicates the methods and models for clustering and distribution fitting analysis. Section 3 describes the data preparation and description for clustering. Sections 4 and 5 present the results of the clustering and distribution fitting analysis, respectively. Finally, Section 6 provides the summary, main results, contributions, and limitations of this paper

## 2. Methodology

This section first describes the overall logic and technology pathway of the study. In the following two subsections, the algorithm and validation for data clustering and the models and test methods for speed distribution fitting are indicated briefly.

## 2.1. Study Logic and Technology Pathway

### 2.1.1. Study Logic

Speed distribution is necessary in traffic simulations and theoretical derivations because speed is a fundamental measure of traffic performance. Detailed and fine simulation and theoretical models take into account individual vehicles with their own demands, preferences, and behavior. To achieve this, it needs the support of speed distributions considering the heterogeneity of cyclists, vehicles and roads, especially the dynamics of subgroups in population. In previous studies, researchers generally adopted a top-down manner which first obtains the speed distributions of bicycle flow and explores the underlying factors further. In this case, it is difficult to identify the natural subgroups because all the potential factors are mixed in the modelling process. Thus, in order to better distinguish the natural subpopulations formed by combinations of factors, a bottom-up approach was designed for the study. The influential factors were discussed firstly, and then the subgroups in population were identified easily by combining the factors or variables. However, it presented too many original subcategories due to the combinations of factor levels or variables. It was not realistic to investigate the speed of so many subcategories and to establish separate model for each. To solve this problem, multi-comparison was used to reduce the levels under each factor and the initial subgroups were further merged using clustering techniques.

Cluster analysis is the organization of a collection of patterns into clusters based on similarity. Fraley and Raftery [10] suggested dividing the clustering approaches into two different groups: hierarchical and partitioning techniques. Han and Kamber [11] suggested the following three additional categories for applying clustering techniques: density-based methods, model-based methods and grid-based methods. An alternative categorization based on the induction principle of different clustering approaches is presented by Castro et al. [12].

Among the partitioning techniques, *K*-means and expectation maximization are two common algorithms. The *K*-means algorithm is the 2nd dominantly used data mining algorithm and the EM algorithm is the 5th dominantly used data mining algorithm [13–15]. The *K*-Means algorithm is a very popular algorithm for data clustering, which aims at the local minimum of the distortion [16,17]. EM is one of the most popular algorithms for statistical pattern recognition and has been widely applied for different purpose: parameter estimation [18,19], mixture simplification [20], image matching [21], and audio-visual scene analysis [22]. EM's popularity has risen due to its use in estimating mixture-model parameters [23,24]. The use of estimated mixture-models is equally interesting for density-estimation tasks [20,25] and clustering tasks [26–28]. For clustering, EM aims at finding clusters such that maximum likelihood of each cluster's parameters are obtained. The two have been compared by many scholars using the datasets from different fields [29–31]. In sum, although EM algorithm produced exceptionally good results in some datasets [32–34], *K*-means has better clustering fitness than EM algorithm considering performance in time complexity and the influence of data type, size, and number of clusters. Most importantly, EM is a model based approach which is based on the assumption that the data are generated by a mixture of underlying probability distributions (commonly Gaussian distribution). In the study, the distribution model remained to be determined in the next step after clustering. Thus, such methods like EM and other mix probability distribution separation are not suitable for our study. Considering the factors above, *K*-means was finally selected as the clustering technique for the study.

### 2.1.2. Technology Pathway

The technology pathway of this study mainly includes four procedures: field data collection and extraction, data prepossessing, data clustering, and speed distribution fitting. The details of the first procedure had been indicated in another paper [35], which conducted an influence factor analysis on bicycle free flow speed and merged the subgroups or levels under each factor as much as possible. Thus, the first procedure is briefly described

herein. In data collection, we used cameras to capture bike traffic operations within 50 m-long track segments marked with some ground tapes and traffic cones. Then, the characteristics of cyclists and vehicles were extracted through manual identification and recording. Afterwards, riding speed was calculated based on the moments that bikes arriving at the marking points.

All data collected are shown in Figure 1 to evaluate the bimodality. The speed distribution does not exhibit bimodality or multimodality, as indicated by a bimodality coefficient 0.429 using the equation in [36]. Therefore, there is no need to use mixture distribution models [37]. As we had examined the influence factors of bicycle speed and the corresponding effects, the exploratory analysis was directly performed on the sub-populations generated from these significant factors to reduce the levels under each factor. The result is shown in Table 1. To address too many sub-populations, clustering was applied and then unimodal distribution models were fitted to the clustering results. The methods and outputs of the remaining three procedures are shown in Figure 2.

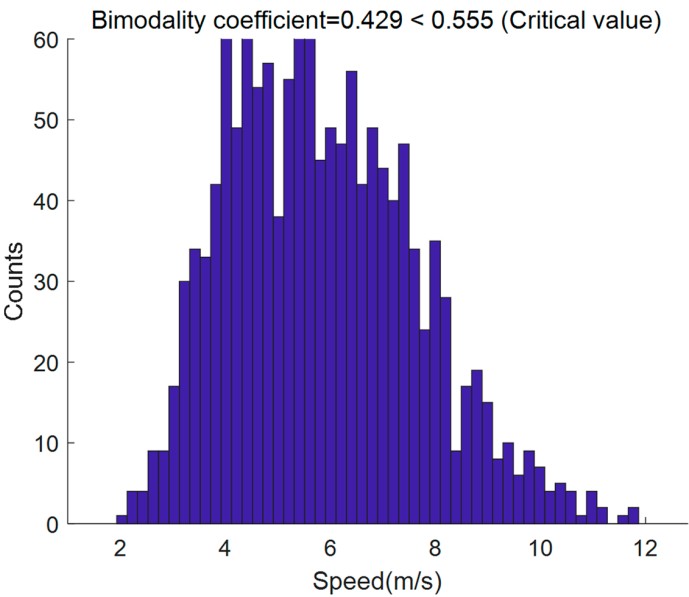

**Figure 1.** Histograms for Speed Data and the bimodality coefficient (value exceeding 0.555 are taken to indicate bimodality and conversely not).

**Table 1.** Basic characteristics of the original and level-merged field survey samples.

| Factor | Original | | | Merged | | |
|---|---|---|---|---|---|---|
| | Category or Level | Counts | Ratio | Category or Level | Counts | Ratio |
| gender | Male | 850 | 62.0% | Male | 850 | 62.0% |
| | Female | 520 | 38.0% | Female | 520 | 38.0% |
| age | (~, 20) years | 42 | 3.0% | (~, 40) years | 830 | 67.9% |
| | (20, 30) years | 308 | 22.5% | | | |
| | (30, 40) years | 580 | 42.4% | | | |
| | (40, 50) years | 296 | 21.6% | (40, 60) years | 440 | 32.1% |
| | (50, 60) years | 144 | 10.5% | | | |
| bicycle type | EB | 1028 | 75.1% | EB | 1028 | 75.1% |
| | CB | 342 | 24.9% | CB | 342 | 24.9% |
| lane width | 2 m | 197 | 14.4% | ≤3.5 m | 367 | 26.8% |
| | 3.4 m | 170 | 12.4% | | | |
| | 3.85 m | 302 | 22.0% | | | |
| | 4 m | 346 | 25.3% | >3.5 m | 1003 | 73.2% |
| | 5 m | 355 | 25.9% | | | |

**Table 1.** *Cont.*

| Factor | Original | | | Merged | | |
|---|---|---|---|---|---|---|
| | Category or Level | Counts | Ratio | Category or Level | Counts | Ratio |
| lateral position | left | 211 | 15.4% | left | 211 | 15.4% |
| | center | 745 | 54.4% | center | 745 | 54.4% |
| | right | 421 | 30.8% | right | 421 | 30.8% |
| Total | ~ | 1370 | ~ | ~ | 1370 | ~ |

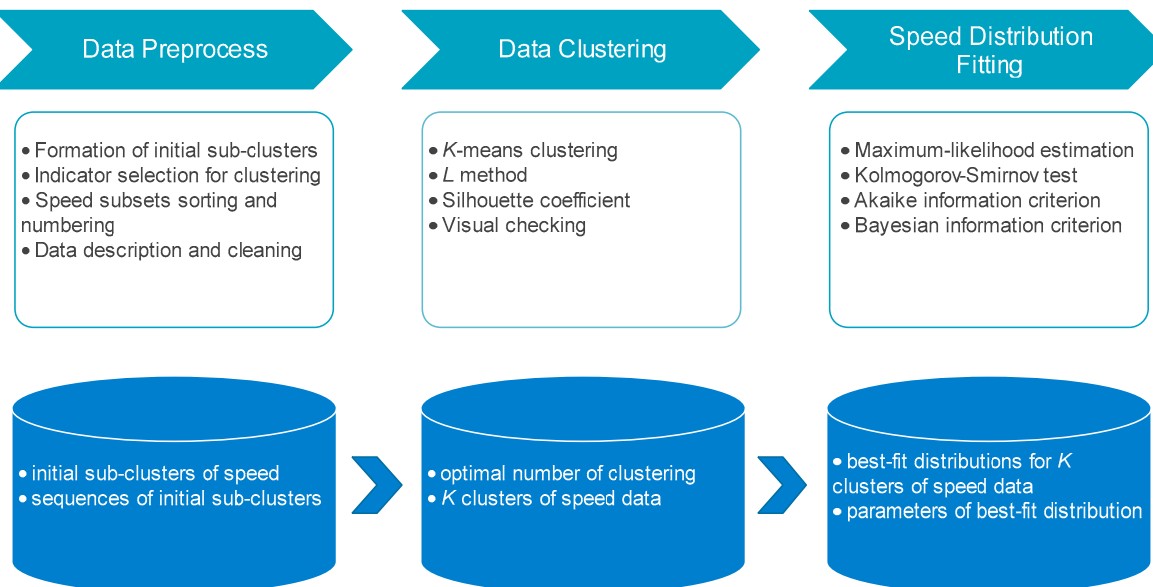

**Figure 2.** Technology path of this study.

## 2.2. Clustering Algorithm and Validation

### 2.2.1. K-Means Clustering

*K*-means clustering is the most widely used partitional clustering algorithm [38]. The goal of *K*-means clustering is to partition n points (which can be one observation or one instance of a sample) into *K* clusters such that each point is assigned to one cluster of which centroid is the closest to it based on a particular proximity measure chosen. The following is an outline of the basic *K*-means algorithm:

Step 1: Select *K* points as initial centroids.

Step 2: Form *K* clusters by assigning each point to its closest centroid.

Step 3: Recompute the centroid of each cluster.

Step 4: Repeat Steps 2–3 until convergence criterion is met.

In the third step, a wide range of proximity measures can be used while computing the closest centroid. The choice can significantly affect the centroid assignment and the quality of the final solution. The different kinds of measures which can be used here are city-block distance, Euclidean distance, correlation distance, and cosine similarity.

### 2.2.2. Determining the Optimal Number of Clusters

The problem of estimating the correct number of clusters (*K*) is one of the major challenges for the *K*-means clustering. The problem can be formulated as how to determine a number of clusters under which each class of data has optimal cohesion and different classes of data have maximum separation, here using an evaluation graph-based *L* method [39].

The information required to determine an appropriate number of clusters/segments to return is contained in an evaluation graph that is created by the clustering/segmentation algorithm. The evaluation graph is a two-dimensional plot where the *x*-axis is the number of clusters, and the *y*-axis is a measure of the quality or error of a clustering consisting of *x* clusters. The *y*-axis values in the evaluation graph can be any evaluation metric, such as: distance, similarity, error, or quality.

Figure 3 shows an example of an evaluation graph, which can be seen in Figure 3a three areas of significantly different data points: a steeply straight region on the left, a flat region on the right, and a gradient region in the middle. On the right side of the Figure 3b, the clustering merging process starts with the smaller classes, with many similar classes being merged, and this trend extends along a straight line to the left. Many of the cluster classes in this region are similar to each other, and so deserve to be merged. Another obvious area of the graph is the rapid increase in merge distance near the left side of the *y*-axis. The rapid increase in merge distance indicates that many different classes are being merged together and that the quality of the clustering is deteriorating because the classes are no longer internally homogeneous. If the merge quality of existing classes starts to get progressively worse, it means that too many mergers have been performed. Therefore a reasonable number of clusters should exist in the asymptotic region of the evaluation chart, or the "*knee*" of the scattered distribution. This area is between the flat area on the right, which increases slowly, and the steep area on the left, which increases rapidly. The number of clusters at the "*knee*" contains a good balance between homogeneity within classes and differences between classes.

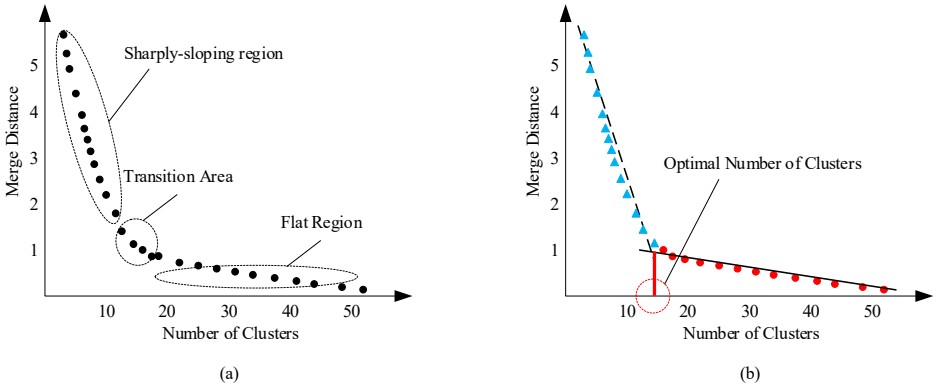

**Figure 3.** (**a**) A sample evaluation graph; (**b**) finding the number of clusters using *L* method.

The problem shifts to finding some point in the asymptotic region (i.e., the best clustering number c), noting that the asymptotic region is between 2 linear regions and that the slope of the tangent at that point is mutated, so it is only necessary to find some point in the asymptotic region and use that point as a boundary to perform a linear fit to the data points in the steep and straight region on the left and the data points in the flat region, and the optimal clustering number is obtained when the fit to the data points in both regions is best at the same time. Now define a metric that captures the interpolated mean of the mixed root mean squared error of the two region fits, as in Equation (1) [39].

$$RMSE_e = \frac{c-1}{b-1}RMSE_c(L_c) + \frac{b-c}{b-1}RMSE_e(R_c) \tag{1}$$

where *RMSE(Lc)* is the root mean squared error of the best-fit line for the sequence of points in *Lc* (and similarly for *Rc*). The weights are proportional to the lengths of *Lc* (*c*–1) and *Rc* (*b*–*c*). *b* is the maximum pre-set number of clusters, i.e., the maxima of the *x*-axis. We seek the value of *c*, such that RMSE is minimized, that is

$$c^{\wedge} = \underset{c}{\mathrm{argmin}} RMSE_c \tag{2}$$

Here, there is a need to specify a range or set for values of $K$ or specifically to set a maximum for $K$. This directly affects the workload and time of clustering. Obviously, the maximum $K$ is far less than the number of initial classes while it should include the optimal value, $c$. By the previous studies summarized by Pham et al. [40], the maximum values of $K$ were usually selected manually and no specific principles or rules were found. However, through calculating the ratio between the maximum number of clusters and the initial number (Maximum $K/N$, %), it was found most were below 20%.

### 2.2.3. Clustering Validation

The silhouette coefficient is a good indicator of the quality of clustering, which combines the cohesion and separation of clusters [38]. The value is in the range of -1 to 1, the larger the value, the better the clustering effect. The specific calculation process is as follows.

(1) For the $i$th element $x_i$, the average of the distances from all other elements in the same cluster as it, recorded as $m_i$, is calculated and used to quantify the cohesion within the cluster.

(2) Select a cluster $n$ other than $x_i$, calculate the average distance between $x_i$ and all points in $n$, iterate through all other clusters to find this nearest average distance, denoted as $n_i$, and use it to quantify the degree of separation between clusters.

(3) For element $x_i$, the profile factor $S_i = (n_i - m_i)/\max(m_i, n_i)$.

(4) Calculate the silhouette coefficients of all $X$'s, the average of which is the overall silhouette coefficient of the current cluster.

To check the clustering result visually, multidimensional scaling (MDS) was applied in the study because of the multidimensionality of the clustering indicators. MDS is an analysis method that maps similarities/differences between multiple objects to points in a low-dimensional space (two-dimensional for example), while maintaining the original relationship between the objects. In this study, multiple statistics characterizing subclass velocities were mapped into two-dimensional space. Then, the results were plotted as scatterplots and examined the goodness of clustering.

### 2.3. Distribution Fitting

#### 2.3.1. Probability Distribution Models for Fitting

15 common continuous probability distributions were considered for fitting the bicycle speed data, including Birnbaum–Saunders, exponential, gamma, generalized extreme value (GEV), generalized Pareto (GP), inverse Gaussian, logistic, loglogistic, lognormal, Nakagami, normal, Rayleigh, Rician, t-location scale, uniform, the mathematical expressions for the probability density functions of these distributions can be found in Table 1 in Appendix A. For each cluster, we found the best-fit parameters for each of these probability distributions using maximum-likelihood estimation. To select the best-fit distribution for a cluster, we then applied multiple model comparison techniques to rank each distribution for every cluster. The subsection below describes these techniques including Kolmogorov–Smirnov test, Akaike information criterion, and Bayesian information criterion.

#### 2.3.2. Model Test and Selection

Kolmogorov–Smirnov Test

The Kolmogorov–Smirnov (K-S) test is a nonparametric test used to decide if a sample is selected from a population with a specific distribution [40]. The K-S test is based on the maximum distance (or supremum) between the empirical distribution function and the normal cumulative distributive function. The Kolmogorov–Smirnov test statistic is defined as:

$$\max_{1 \leq i \leq N} \left( F(y_i) - \frac{i-1}{N}, \frac{i}{N} - F(y_i) \right) \tag{3}$$

where $F$ is the cumulative distribution function of the samples being tested. If the probability that a given value of $D$ is very small (less than a certain critical value, which can be

obtained from tables), we can reject the null hypothesis that the two samples are drawn from the same underlying distributions at a given confidence level.

AIC and BIC

The Akaike information criterion (AIC) is a way of selecting a model from an input set of models [41,42]. It can be derived by an approximate minimization of the Kullback-Leibler distance between the model and the truth. It is based on information theory, but a heuristic way to think about it is as a criterion that seeks a model, which has a good fit to the truth with very few parameters.

It is defined as [41]:

$$AIC = -2LL + 2k \tag{4}$$

where $LL$ is the log-likelihood of the model on the dataset, and $k$ is the number of parameters in the model.

The bias-corrected information criterion, often called AICc, takes into account the finite sample size, by essentially increasing the relative penalty for model complexity with small datasets.

It is defined as [42]:

$$AICc = AIC + \frac{2k(k+1)}{N-k-1} \tag{5}$$

where $N$ is the sample size. For this study, we have used AICc for evaluating model efficacy.

Bayesian information criterion (BIC) is also an alternative way of selecting a model from a set of models. It is an approximation to Bayes factor between two models. It is defined as [41]:

$$BIC = -2LL + k\log(N) \tag{6}$$

When comparing the BIC values for two models, the model with the smaller BIC value is considered better. In general, BIC penalizes models with more parameters more than AICc does.

## 3. Data Preparation and Description

This section indicates data preparation process for clustering including formation of initial sub-clusters of speed data, indicator selection for clustering, speed subsets sorting. Moreover, the speed data of initial sub-clusters are described in the end.

Formation of initial sub-clusters: our previous study had determined five significant factors, including gender, age, bicycle type, lane width, lateral position, impacting BFFS. The values or levels of these factors had been merged as far as possible. By comparative analysis, bicycle speed has two distinct categories for gender (male and female), age ($\leq$40 years and >40 years), bicycle type (conventional bicycle (CB) and electric bicycle (EB)) and lane width ($\leq$3.5 m and >3.5 m) respectively. In respect to lateral position, the speed data collected from the three parts of a bicycle lane (left, center and right) are significantly different. In the study, the five factors mentioned above were considered as the categorical variables, of which combinations partitioned the bicycle speed data collected into the initial sub-clusters or subsets. Theoretically, there are 48 ($2 \times 2 \times 2 \times 2 \times 3$) combinations for the five categorical variables and therefore exists 48 initial speed subsets. However, field observation only captured 45 sub-clusters and the unobserved three were sub-cluster (female, >40 years, EB, $\leq$3.5 m, left), (female, >40 years, CB, $\leq$3.5 m, left), and (male, $\leq$40 years, CB, $\leq$3.5 m, left).

Indicator selection for clustering: Further clustering on the basis of the 45 initial sub-clusters was first necessary to determine the indicators for clustering. Mean, standard deviation, minimum and maximum were selected to characterize the speed data of the sub-clusters. These four statistics measured the central tendency, variability, and the lower and upper boundaries of the variation, respectively, which can completely describe the distribution of bicycle velocity of a subclass.

Speed subsets sorting and numbering: After determining the indicators for clustering, sorting and numbering the speed subsets were performed to obtain the continuity of indicator changes, which was helpful to detect and correct the data defects in speed data and indicators. Moreover, the gradual variability of the indicators was also reflected in the final clustering results, in which the subset numbers in each cluster were orderly sequential. This can confirm the validity of the clustering results from one side. Therefore, the speed subsets (indicator: means) were multiple-level sorted in ascending order based on the degree of influence of the categorical variables on bicycle velocity, which had been examined by our preliminary study [24]. Those results indicated that bicycle type (CB < EB in speed) was the most influential factor on BFFS, followed by bicycle lateral position (right < center < left), gender (female < male), age (the older (>40 years) < the young (≤40 years)), and lane width (the narrow < the wide) in sequence. The sorting result is shown in Table A2.

Data description and cleaning of speed subsets: When completing data preparation, speed subsets are plotted in boxplot as shown in Figure 4 in order to check the outliers (sub-cluster 8), the abnormal (sub-cluster 17), incomplete data (sub-cluster 20) and other data defects. To address the defects above, data cleaning techniques were applied, mainly including filtering out outliers, fixing the abnormal and impute missing values.

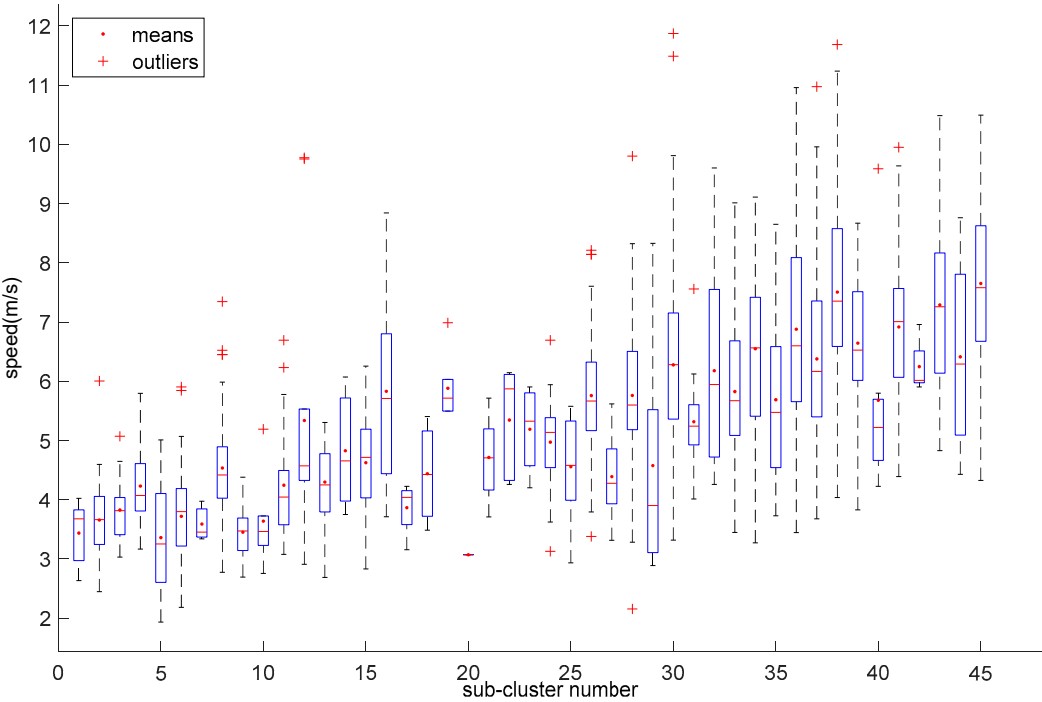

**Figure 4.** Data Description for 45 Speed Subsets.

## 4. Clustering Results

### 4.1. Optimal Number of Clusters

Integrating the previous studies and the initial number of the sub-clusters, we selected 15 as the maximum $K$ (at this time maximum $K/N$ = 33% > 20% before), and then 2–15 were the pre-set number of clusters. The four common methods of calculating the new clustering centroids were adopted, and finally, the clustering quality of the methods was compared to determine the best one. The metric used in $L$ method was the means of the distance from each data point to the centroid of its cluster. The calculation result of the $L$ method is shown in Figure 5.

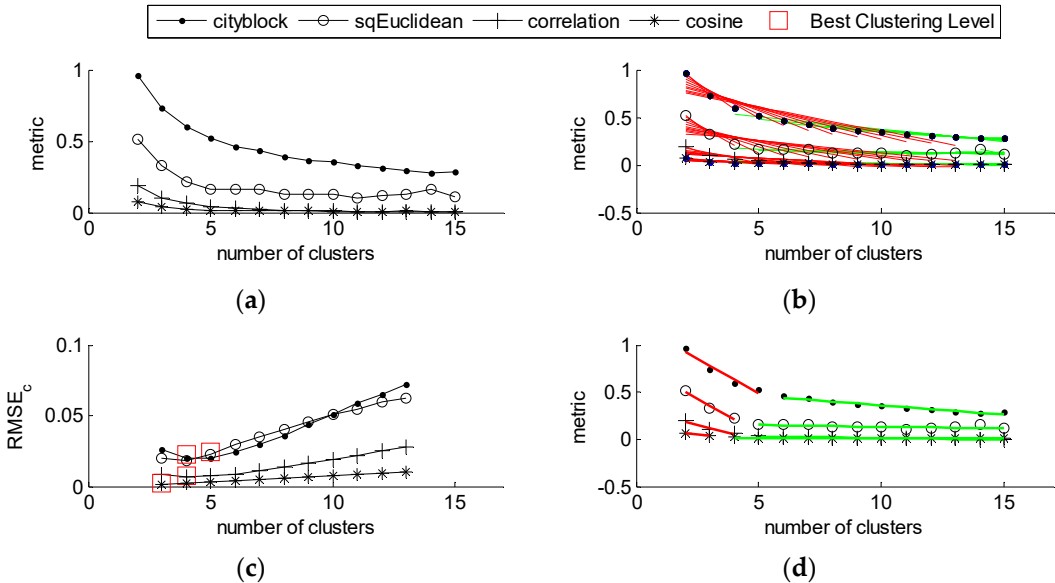

**Figure 5.** (**a**) evaluation graph, (**b**) possible fitting lines, (**c**) RMSE, and (**d**) Best-fit lines for city-block distance, Euclidean distance, correlation distance, and Cosine similarity.

Figure 5 obviously indicates that cosine similarity presents the best performance in clustering evaluation. For each pre-set $K$ number, the value of cosine similarity is the least of four methods, which represents the best cohesion of clusters. Meanwhile, cosine similarity renders the least number of cluster (3 clusters) when the others obtain 4 or 5.

### 4.2. Validaty of Clustering

Silhouette coefficient and two-dimensional graph were applied to evaluate the quality of clustering, as shown in Figure 6. From Figure 6a, it can be seen that the silhouette coefficients of most sub-clusters under each cluster exceeds 0.6 (basic criterion), and most of them exceed 0.8. Thus, for optimal number 3, the clustering presents a good performance.

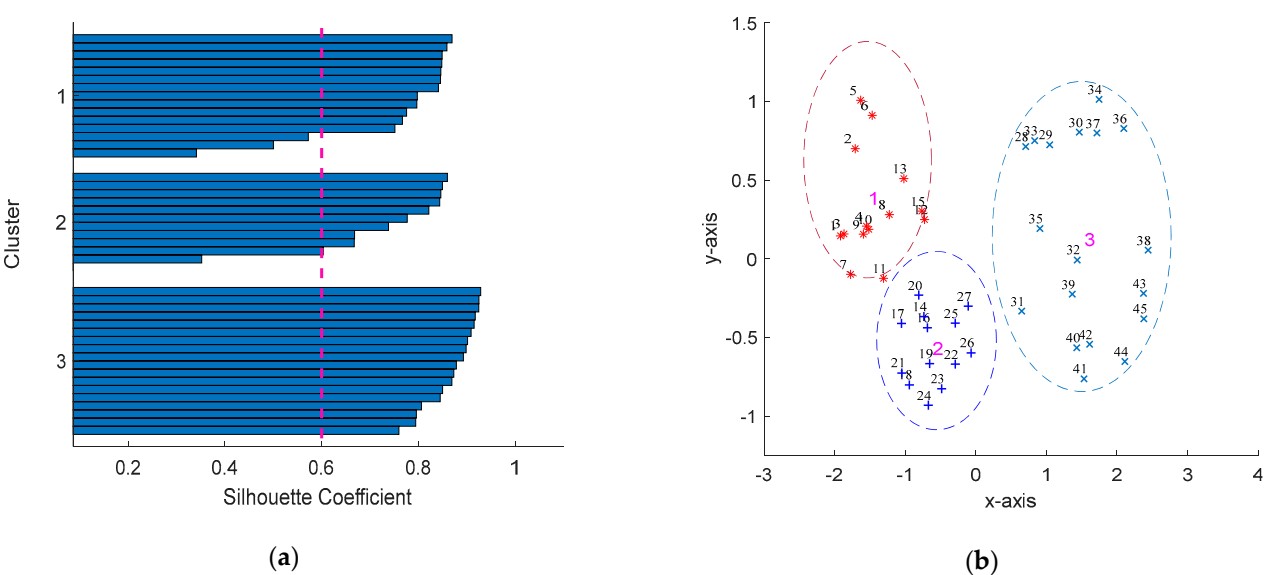

**Figure 6.** (**a**) Silhouette coefficient values, and (**b**) Visualization of clustered data when number of clustering = 3.

Four clustering indicators of sub-clusters were mapped to points in a two-dimensional space see Figure 6b using MDS in order to check the clustering results of $K = 3$. It is obvious that 45 sub-clusters forms three sets, namely Cluster 1, 2, and 3. Simultaneously,

sub-cluster numbers are serial in each cluster, sub-cluster 1–15, 16–27, and 28–45 for Cluster 1, 2, 3, respectively, which also conforms the validity of the clustering result as indicated in 3.1. Moreover, Xu et al. also obtained the optimal number 3 in modelling bicycle free flow speed [6].

To address the three sub-categories unobserved, they were imputed according to the aforementioned sorting rules. The three were numbered 19, 23, and 41 respectively. Among them, sub-cluster 19 and 23 were categorized to Cluster 2 while sub-cluster 41 to Cluster 3.

## 5. Distribution Fitting Results for Speed Clusters

### 5.1. Speed Distribution of Clusters

The statistics and distribution characteristics of the three clusters were calculated, as shown in Table 2. Three speed subsets do not presents the bimodality or multimodality with their bimodality coefficients below 0.555. Thus, 15 common statistical distributions listed above were applied to fit the speed data of the three clusters and the test statistics for K-S test, AIC, AICc, and BIC were computed for the probability distributions. For each cluster, we ranked all the probability distribution functions, using each of the four model test techniques in decreasing order as shown in Tables A3–A5 in Appendix B and the eight best-fit functions for each cluster are show in Figure 7a–f. Table 3 summarizes the distribution orders for the three clusters and sorts these distributions by descending order on sum of three rankings and variances.

**Table 2.** Speed statistics and distribution features for the three cluters.

| Cluster | Counts | Speed Statistics (m/s) | | | | | | Distribution Features | | |
|---------|--------|--------|------|-------|--------------|-------|-------|----------|----------|------|
| | | Median | Mean | Std * | 85th Value | Min * | Max * | Kurtosis | Skewness | BC * |
| 1 | 327 | 4.00 | 4.06 | 0.95 | 4.81 | 1.93 | 9.77 | 10.37 | 1.63 | 0.35 |
| 2 | 179 | 5.25 | 5.24 | 1.10 | 6.32 | 2.83 | 8.84 | 3.29 | 0.34 | 0.33 |
| 3 | 864 | 6.54 | 6.65 | 1.58 | 8.22 | 2.16 | 11.87 | 3.03 | 0.40 | 0.38 |
| Overall | 1370 | 5.66 | 5.85 | 1.78 | 7.73 | 1.93 | 11.87 | 2.85 | 0.48 | 0.43 |

\* Std = Standard error, Min = Minimum, Max = Maximum, BC = bimodality coefficient.

**Table 3.** Goodness-of-Fit Rankings of 15 Distribution Models for the Three Clusters.

| Number | Name | Cluster 1 | Cluster 2 | Cluster 3 | Sum | Var. | Suggestions |
|--------|------|-----------|-----------|-----------|-----|------|-------------|
| 1 | Gev | 6 | 3 | 1 | 10 | 4.22 | Recommended |
| 2 | Gamma | 8 | 1 | 2 | 11 | 9.56 | Recommended |
| 3 | Lognormal | 3 | 4 | 6 | 13 | 1.56 | Recommended |
| 4 | Birnbaumsaunders | 5 | 5 | 4 | 14 | 0.22 | Suitable |
| 5 | Inversegaussian | 4 | 7 | 5 | 16 | 1.56 | Suitable |
| 6 | Loglogistic | 2 | 6 | 9 | 17 | 8.22 | Suitable |
| 7 | Tlocationscale | 1 | 11 | 10 | 22 | 20.22 | Suitable |
| 8 | Nakagami | 9 | 2 | 3 | 14 | 9.56 | Uncertain |
| 9 | Rician | 10 | 8 | 7 | 25 | 1.56 | Uncertain |
| 10 | Normal | 11 | 9 | 8 | 28 | 1.56 | Uncertain |
| 11 | Logistic | 7 | 10 | 11 | 28 | 2.89 | Uncertain |
| 12 | Uniform | 13 | 12 | 12 | 37 | 0.22 | Unsuitable |
| 13 | Rayleigh | 12 | 13 | 13 | 38 | 0.22 | Unsuitable |
| 14 | GP | 14 | 14 | 14 | 42 | 0.00 | Unsuitable |
| 15 | Exponential | 15 | 15 | 15 | 45 | 0.00 | Unsuitable |

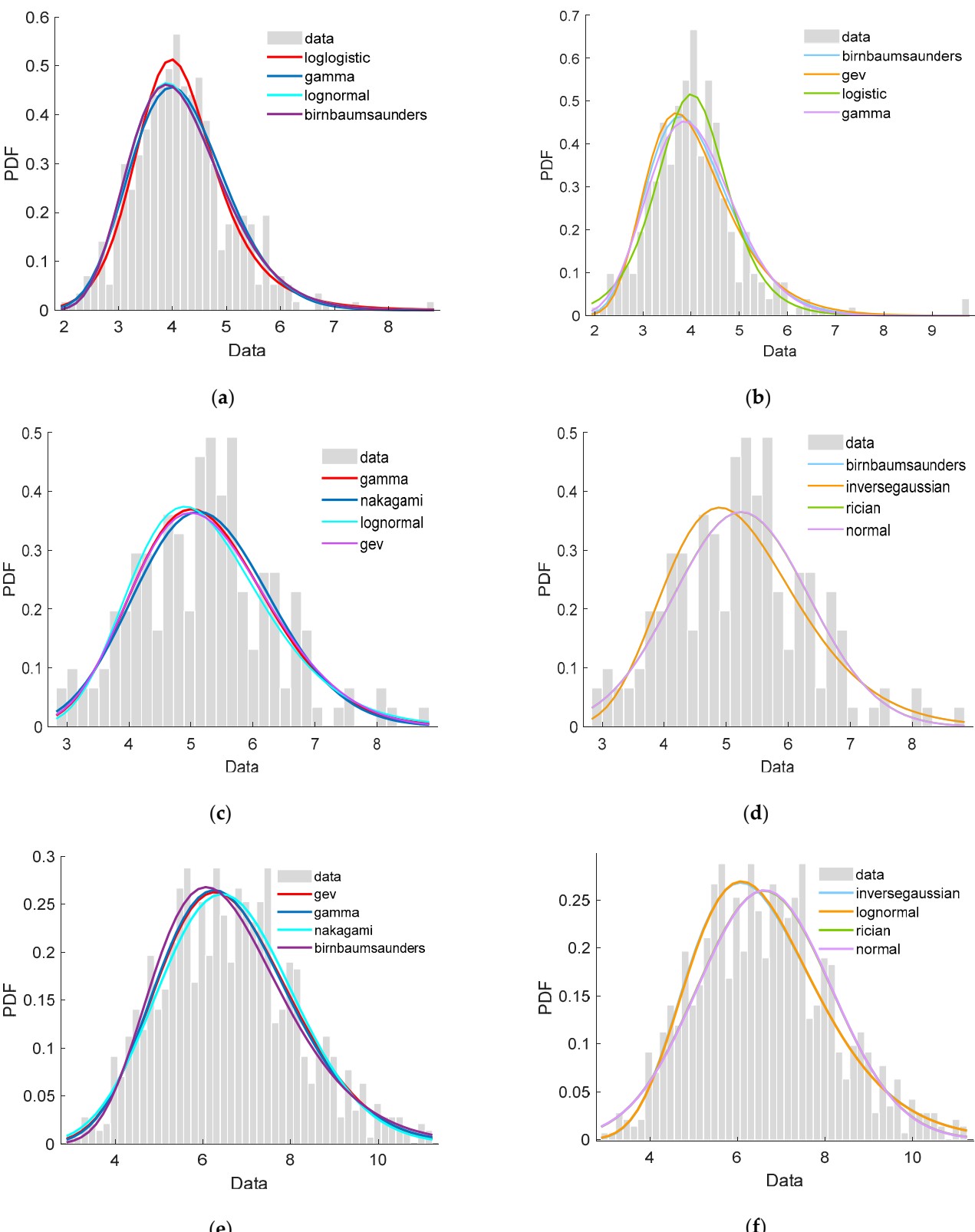

**Figure 7.** (**a**) Fitting results for models ranked in the 1–4, (**b**) and 5–8 for the Cluster 1; (**c**) Fitting results for models ranked in the 1–4, (**d**) and 5–8 for the Cluster 2; (**e**) Fitting results for models ranked in the 1–4, (**f**) and 5–8 for the Cluster 3.

Our results from each of the model comparison tests are summarized as follows:

- The first 7 distributions were suitable to fit the data of the three clusters according to the K-S test results while Uniform, Rayleigh, Gp, and exponential distribution were not. Nakagami, Rician, normal, logistic remained uncertain due to the failures of passing at least one of K-S tests to the three clusters.
- After considering the sum and variance of the three rankings together, we recommended GEV, Gamma, and Lognormal distributions as the top three tools to fit the three clusters of speed data set.
- Moreover, Tlocationscale, Gamma, and GEV distributions performed best in fitting the data from Clusterss 1, 2, and 3, respectively.

*5.2. Discussion on Best-Fit Distribution*

Among the three recommended distributions, GEV and Gamma distributions provided the best fit to the speed data. GEV presented a relatively better stability than Gamma and an increasingly optimizing performance from Cluster 1 to 3.

The GEV distribution is often used to model the smallest or largest value among a large set of independent, identically distributed random values representing measurements or observations [43]. It combines three simpler distributions into a single form, allowing a continuous range of possible shapes that include all three of the simpler distributions. The three distribution types correspond to the limiting distribution of block maxima from different classes of underlying distributions illustrated in Figure 8:

- Type I—Distributions whose tails decrease exponentially when the shape parameter ($k$) is equal to zero, see the light blue line.
- Type II—Distributions whose tails decrease as a polynomial shown by the yellow line, when $k$ is more than zero.
- Type III—Distributions whose tails are finite as illustrated by the red line, when $k$ is less than zero.

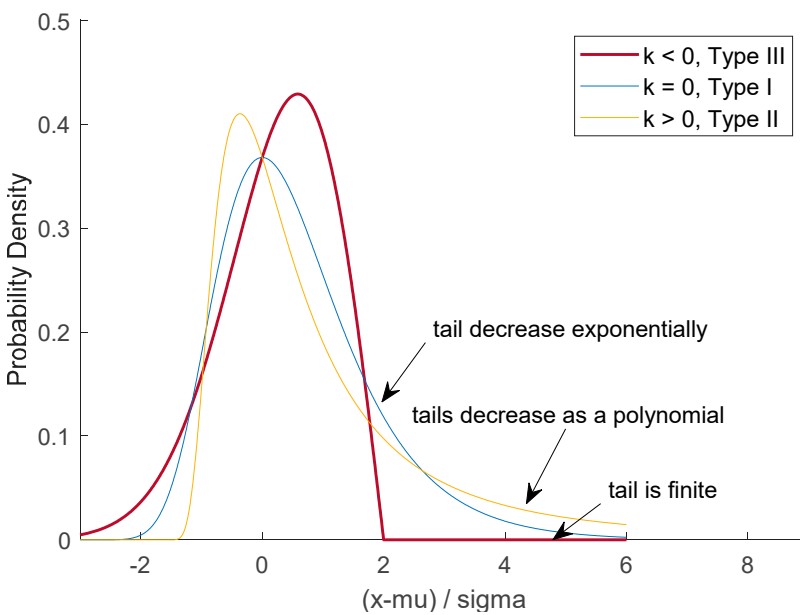

**Figure 8.** An example for three types of extreme value distributions and their tails' features.

The free speed is the maximum speed a bicycle can achieve most of the time, consistent with the case where the GEV distribution applies. The best performance of GEV in fitting to the speed dataset of the three clusters had proved this point. Moreover, the estimated shape parameters ($k$) were all less than zero see Tables A3–A5, which referred to Type III of the extreme value distribution. It means that bicycles have their speed limits due to the source of driving, no matter powered by physical strength or electric power. From another side, Type III of the extreme value distribution is also applied to establish a speed

model with a safety or maximum velocity required by transport laws or regulations. From 2002 onwards, European Union had been published a series of directives or laws which defined the maximum velocity of an e-bike as 25 km/h (15.5 mile/h) [44,45]. This limit was then adopted by many countries in Europe [46,47] and other regions like Victoria in Australia [48]. In 2018, China updated the design speed of EB in safety technical specification for electric bicycle [49]. This code also referred the requirements of EU and regulates 25 km/h as the maximum speed of electric bicycles produced from 15 April 2019. Therefore, it is necessary to update the distribution model of bicycle free flow speed under the limit.

### 6. Conclusions

The paper first performed *K* means clustering analysis on the speed sub-clusters formed by the combinations of five categorical variables (bicycle type, bicycle lateral position, gender, age and lane width) and determined the optimal number of clusters using *L* method. The clustering results were then validated and checked with silhouette coefficient and two-dimensional graph. Afterwards, 15 common models were fitted to the three clusters of speed data and four model test technologies were applied to select the best-fit distribution. The following results and conclusions were acquired:

- 48 initial bicycle speed sub-clusters generated by the combinations of bicycle type, bicycle lateral position, gender, age, and lane width were grouped in three clusters finally.
- Among the common distributions, GEV, Gamma, and lognormal were the top three models to fit the three clusters of speed dataset.
- Integrating stability and overall performance, GEV was the best-fit distribution of bicycle speed. The speeds of the three clusters followed GEV $(-0.04, 0.78, 3.66)$, GEV $(-0.17, 1.03, 4.81)$, and GEV $(-0.18, 1.42, 6.00)$, respectively.

The conclusions of the paper provides a useful reference for researchers when they select a suitable model to describe bicycle speed in other contexts. In simulations and theoretical derivations, the best-fit models found in the study may be considered as the alternative tools. Moreover, the fitting results are applied by finding the corresponding cluster if it just happens to match. Although the suitability of the models and best-fit distributions were validated, more data and many other contexts are necessary to improve the performance of models. Specifically, gamma distribution performed better or similarly than GEV. The model deserves more discussions with the better quality of data. We will continue to work on the improvement of bicycle speed models in the future.

Moreover, due to the limitations of the data extraction technology at the time of this study, the demographic attributes of cyclists like age and gender were estimated manually, which lowers the accuracy of the models to some extent. These limitations will be improved with new AI technologies like object and character recognition in our next studies.

**Author Contributions:** Z.Y. undertook the data collection. X.Y. (Xingchen Yan) provided an interpretation of the results and wrote the majority of the paper. T.W., X.Y. (Xiaofei Ye), and H.B. contributed to the paper review and editing. J.C. was the supervisor of the paper. All authors have read and agreed to the published version of the manuscript.

**Funding:** This research was funded by the National Key R&D Program of China (grant No. 2018YFC0 704704), Natural Science Foundation of Jiangsu Province (grant No. BK20180775&BK20170932), Key Project of National Natural Science Foundation of China (grant No. 51638004), Natural Science Foundation of Zhejiang Province (LY20E080011), Fund for Less Developed Regions of the National Natural Science Foundation of China (grant No. 71861006), Natural Science Foundation of Guangxi Province (2019JJA160194), Guangxi Science and Technology Base and Talent Special Program (2019AC20137), Basic public welfare research project of Zhejiang Province 2018(LGF18E090005), and National Key Research and Development Program of China (2017YFE9134700).

**Acknowledgments:** The authors would like to express their sincere thanks to the anonymous reviewers for their constructive comments on an earlier version of this manuscript.

**Conflicts of Interest:** The authors declare no conflict of interest.

## Appendix A

**Table 1.** Probability density functions of different distributions used to fit the speed data.

| Distribution | Probability Density Function | Parameters |
|---|---|---|
| Birnbaumsaunders | $\left(\frac{\sqrt{\frac{x-\mu}{\beta}}+\sqrt{\frac{\beta}{x-\mu}}}{2\gamma(x-\mu)}\right)\phi\left(\frac{\sqrt{\frac{x-\mu}{\beta}}+\sqrt{\frac{\beta}{x-\mu}}}{\gamma}\right); x > \mu$ | $\beta$: scale parameter, $\beta > 0$; <br> $\gamma$: shape parameter, $\gamma > 0$. |
| Ev | $\frac{1}{\beta}e^{-\frac{x-\mu}{\beta}}e^{-e^{\frac{x-\mu}{\beta}}}$ | $\mu$: location parameter; <br> $\sigma$: scale parameter, $\sigma \geq 0$ |
| Exponential | $\frac{1}{\theta}e^{-\frac{x}{\theta}}; x \geq 0$ | $\theta$: inverse scale, $\theta > 0$ |
| Gamma | $\frac{1}{\Gamma(\alpha)\beta^\alpha}x^{\alpha-1}e^{-\frac{x}{\beta}}; x > 0$ | $\alpha$: shape parameter, $\alpha: > 0$; <br> $\beta$: scale paramete, $\beta > 0$ |
| Gev | $\frac{1}{\sigma}t(x)^{k+1}e^{-t(x)}; t(x) = \begin{cases} \left(1+k\left(\frac{x-\theta}{\sigma}\right)\right), k \neq 0 \\ e^{-\frac{x-\theta}{\sigma}}, k = 0 \end{cases}$ | $k$: shape parameter <br> $\sigma$: scale parameter, $\sigma > 0$; <br> $\theta$: location parameter |
| Gp | $\begin{cases} \frac{1}{\sigma}\left(1+k\frac{x-\theta}{\sigma}\right)^{-1-\frac{1}{k}}, k \neq 0 \\ \frac{1}{\sigma}e^{-\frac{x-\theta}{\sigma}}, k = 0 \end{cases}$ | $k$: shape parameter <br> $\sigma$: scale parameter, $\sigma \geq 0$; <br> $\theta$: location parameter |
| Inversegaussian | $\sqrt{\frac{\lambda}{2\pi^3}}e^{\frac{-\lambda(x-\mu)^2}{2\mu^2 x}}; x > 0$ | $\mu$: scale parameter, $\mu > 0$; <br> $\lambda$: shape parameter, $\lambda > 0$ |
| Logistic | $\frac{1}{\beta}\frac{e^{-\frac{x-\mu}{\beta}}}{\left[1+e^{-\frac{x-\mu}{\beta}}\right]}$; | $\mu$: mean; <br> $\beta$: scale parameter, $\beta > 0$ |
| Loglogistic | $\frac{1}{\sigma}\frac{1}{x}\frac{e^z}{(1+e^z)^2}; z = \frac{\log x-\mu}{\sigma}, x > 0$ | $\mu$: mean of logarithmic values, $\mu > 0$; <br> $\sigma$: scale parameter of logarithmic values, $\sigma > 0$; |
| Lognormal | $\frac{1}{\sqrt{2\pi}\sigma}\frac{1}{x}e^{-\frac{(\log x-\mu)^2}{2\sigma^2}}; x > 0$ | $\mu$: mean of logarithmic values; <br> $\sigma$: standard deviation of logarithmic values, $\sigma > 0$; |
| Nakagami | $2\left(\frac{\mu}{\omega}\right)^\mu\frac{1}{\Gamma(\mu)}x^{(2\mu-1)}e^{\frac{-\mu}{\omega}x^2}; x > 0$ | $\mu$: shape parameter, $\mu > 0$; <br> $\omega$: scale parameter, $\omega > 0$ |
| Normal | $\frac{1}{\sqrt{2\pi}\sigma}e^{-\frac{(x-\mu)^2}{2\sigma^2}}$; | $\mu$: mean; <br> $\sigma$: standard deviation, $\sigma \geq 0$; |
| Rayleigh | $\frac{x}{b^2}e^{-\frac{x^2}{b^2}}; x > 0$ | $b > 0$ |
| Rician | $l_0\frac{xs}{\sigma^2}\frac{x}{\sigma^2}e^{-\frac{x^2+s^2}{2\sigma^2}}; x > 0$ | $s$: noncentrality parameter, $s \geq 0$; <br> $\sigma$: scale parameter, $\sigma > 0$; |
| Tlocationscale | $\frac{\Gamma\left(\frac{v+1}{2}\right)}{\sigma\sqrt{v\pi}\Gamma\left(\frac{v}{2}\right)}\left(\frac{v+\left(\frac{x-\mu}{\sigma}\right)^2}{v}\right)^{-\frac{v+1}{2}}$ | $\mu$: location parameter; <br> $\sigma$: scale parameter, $\sigma > 0$; <br> $v$: shape parameter, $v > 0$. |
| Uniform | $\frac{1}{b-a}; a \leq x \leq b$ | $a$: lower parameter; <br> $b$: upper parameter |
| Weibull | $\frac{b}{a}x^{a-1}e^{-\frac{xb}{a}}; x > 0$ | $a$: scale parameter, $a > 0$; <br> $b$ scale parameter, $b > 0$; |

Table A2. Sorting and Clustering Results for 48 sub-clusters.

| Number | Gender | Age | Bicycle Type | Lane Width | Lateral Position | Cluster |
|--------|--------|-----|--------------|------------|------------------|---------|
| 1 | Female | >40 years | CB | ≤3.5 m | right | 1 |
| 2 | Female | >40 years | CB | >3.5 m | right | 1 |
| 3 | Female | ≤40 years | CB | ≤3.5 m | right | 1 |
| 4 | Female | ≤40 years | CB | >3.5 m | right | 1 |
| 5 | Male | >40 years | CB | ≤3.5 m | right | 1 |
| 6 | Male | >40 years | CB | >3.5 m | right | 1 |
| 7 | Male | ≤40 years | CB | ≤3.5 m | right | 1 |
| 8 | Male | ≤40 years | CB | >3.5 m | right | 1 |
| 9 | Female | >40 years | CB | ≤3.5 m | center | 1 |
| 10 | Female | >40 years | CB | >3.5 m | center | 1 |
| 11 | Female | ≤40 years | CB | ≤3.5 m | center | 1 |
| 12 | Female | ≤40 years | CB | >3.5 m | center | 1 |
| 13 | Male | >40 years | CB | ≤3.5 m | center | 1 |
| 14 | Male | >40 years | CB | >3.5 m | center | 1 |
| 15 | Male | ≤40 years | CB | ≤3.5 m | center | 1 |
| 16 | Male | ≤40 years | CB | >3.5 m | center | 2 |
| 17 | Female | >40 years | CB | ≤3.5 m | left | 2 |
| 18 | Female | >40 years | CB | >3.5 m | left | 2 |
| 19 | Female | ≤40 years | CB | ≤3.5 m | left | 2 |
| 20 | Female | ≤40 years | CB | >3.5 m | left | 2 |
| 21 | Male | >40 years | CB | ≤3.5 m | left | 2 |
| 22 | Male | >40 years | CB | >3.5 m | left | 2 |
| 23 | Male | ≤40 years | CB | ≤3.5 m | left | 2 |
| 24 | Male | ≤40 years | CB | >3.5 m | left | 2 |
| 25 | Female | >40 years | EB | ≤3.5 m | right | 2 |
| 26 | Female | >40 years | EB | >3.5 m | right | 2 |
| 27 | Female | ≤40 years | EB | ≤3.5 m | right | 2 |
| 28 | Female | ≤40 years | EB | >3.5 m | right | 2 |
| 29 | Male | >40 years | EB | ≤3.5 m | right | 2 |
| 30 | Male | >40 years | EB | >3.5 m | right | 3 |
| 31 | Male | ≤40 years | EB | ≤3.5 m | right | 3 |
| 32 | Male | ≤40 years | EB | >3.5 m | right | 3 |
| 33 | Female | >40 years | EB | ≤3.5 m | center | 3 |
| 34 | Female | >40 years | EB | >3.5 m | center | 3 |
| 35 | Female | ≤40 years | EB | ≤3.5 m | center | 3 |
| 36 | Female | ≤40 years | EB | >3.5 m | center | 3 |
| 37 | Male | >40 years | EB | ≤3.5 m | center | 3 |
| 38 | Male | >40 years | EB | >3.5 m | center | 3 |
| 39 | Male | ≤40 years | EB | ≤3.5 m | center | 3 |
| 40 | Male | ≤40 years | EB | >3.5 m | center | 3 |
| 41 | Female | >40 years | EB | ≤3.5 m | left | 3 |
| 42 | Female | >40 years | EB | >3.5 m | left | 3 |
| 43 | Female | ≤40 years | EB | ≤3.5 m | left | 3 |
| 44 | Female | ≤40 years | EB | >3.5 m | left | 3 |
| 45 | Male | >40 years | EB | ≤3.5 m | left | 3 |
| 46 | Male | >40 years | EB | >3.5 m | left | 3 |
| 47 | Male | ≤40 years | EB | ≤3.5 m | left | 3 |
| 48 | Male | ≤40 years | EB | >3.5 m | left | 3 |

## Appendix B

**Table A3.** Probability Distribution Rankings Using Different Goodness-Of-Fit Tests for Cluster.

| Order | Name | Parameters | LL | KS | AIC | AICc | BIC |
|---|---|---|---|---|---|---|---|
| 1 | loglogistic | $\mu$: 1.38, $\sigma$: 0.12 | −411.6 | Y | 827.3 | 827.3 | 834.9 |
| 2 | tlocationscale | $\mu$: 3.99, $\sigma$: 0.67, $\nu$: 4.17 | −416.8 | Y | 839.7 | 839.8 | 851.1 |
| 3 | lognormal | $\mu$: 1.38, $\sigma$: 0.22 | −419.2 | Y | 842.5 | 842.5 | 850.0 |
| 4 | inversegaussian | $\mu$: 4.06, $\lambda$: 81.49 | −420.2 | Y | 844.4 | 844.5 | 852.0 |
| 5 | birnbaumsaunders | $\beta$: 3.97, $\gamma$: 0.22 | −420.3 | Y | 844.6 | 844.6 | 852.2 |
| 6 | gev | $k$: −0.04, $\sigma$: 0.78, $\theta$: 3.66 | −420.4 | Y | 846.8 | 846.8 | 858.1 |
| 7 | logistic | $\mu$: 4.00, $\beta$: 0.48 | −421.8 | Y | 847.6 | 847.6 | 855.1 |
| 8 | gamma | $\alpha$: 20.44, $\beta$: 0.20 | −423.8 | Y | 851.5 | 851.6 | 859.1 |
| 9 | nakagami | $\mu$: 5.05, $\omega$: 17.42 | −433.7 | N | 871.3 | 871.3 | 878.9 |
| 10 | rician | $s$: 3.95, $\sigma$: 0.96 | −445.3 | N | 894.7 | 894.7 | 902.2 |
| 11 | normal | $\mu$: 4.06, $\sigma$: 0.95 | −446.1 | N | 896.3 | 896.3 | 903.9 |
| 12 | rayleigh | $b$: 2.95 | −584.2 | N | 1170.4 | 1170.5 | 1174.2 |
| 13 | uniform | $a$: 1.93, $b$: 9.77 | −673.4 | N | 1350.8 | 1350.8 | 1358.4 |
| 14 | gp | −0.56, $\theta$: 5.53 | −703.1 | N | 1410.1 | 1410.2 | 1417.7 |
| 15 | exponential | $\theta$: 4.06 | −785.5 | N | 1573.1 | 1573.1 | 1576.9 |

**Table A4.** Probability Distribution Rankings Using Different Goodness-Of-Fit Tests for Cluster.

| Order | Name | Parameter Values | LL | KS | AIC | AICc | BIC |
|---|---|---|---|---|---|---|---|
| 1 | gamma | $\alpha$: 22.74, $\beta$: 0.23 | −268.2 | Y | 540.5 | 540.6 | 546.9 |
| 2 | nakagami | $\mu$: 5.92, $\omega$: 28.66 | −268.3 | Y | 540.7 | 540.7 | 547.0 |
| 3 | gev | $k$: -0.17, $\sigma$: 1.03, $\theta$: 4.81 | −269.5 | Y | 542.9 | 543.0 | 549.3 |
| 4 | lognormal | $\mu$: 1.63, $\sigma$: 0.21 | −268.5 | Y | 542.9 | 543.1 | 552.5 |
| 5 | birnbaumsaunders | $\beta$: 5.12, $\gamma$: 0.21 | −269.5 | Y | 543.0 | 543.0 | 549.3 |
| 6 | tlocationscale | $\mu$: 5.23, $\sigma$: 1.04, $\nu$: 20.78 | −269.5 | Y | 543.1 | 543.1 | 549.4 |
| 7 | inversegaussian | $\mu$: 5.24, $\lambda$: 113.26 | −269.7 | Y | 543.4 | 543.5 | 549.8 |
| 8 | rician | $s$: 5.12, $\sigma$: 1.11 | −269.8 | Y | 543.6 | 543.7 | 550.0 |
| 9 | normal | $\mu$: 5.24, $\sigma$: 1.09 | −270.2 | Y | 544.4 | 544.4 | 550.7 |
| 10 | logistic | $\mu$: 5.22, $\beta$: 0.62 | −270.3 | Y | 544.6 | 544.7 | 551.0 |
| 11 | loglogistic | $\mu$: 1.64, $\sigma$: 0.12 | −269.5 | Y | 545.0 | 545.2 | 554.6 |
| 12 | uniform | $a$: 2.83, $b$: 8.84 | −321.1 | N | 646.1 | 646.2 | 652.5 |
| 13 | rayleigh | $b$: 3.79 | −363.0 | N | 728.1 | 728.1 | 731.2 |
| 14 | gp | $\sigma$: -1.01, $\theta$: 8.93 | −389.9 | N | 783.7 | 783.8 | 790.1 |
| 15 | exponential | $\theta$: 5.24 | −475.5 | N | 953.0 | 953.1 | 956.2 |

**Table A5.** Probability Distribution Rankings Using Different Goodness-Of-Fit Tests for Cluster.

| Order | Name | Parameter Values | LL | KS | AIC | AICc | BIC |
|---|---|---|---|---|---|---|---|
| 1 | gev | $k$: −0.18, $\sigma$:1.42, $\theta$: 6.00 | −1566.6 | Y | 3139.3 | 3139.3 | 3153.5 |
| 2 | gamma | $\alpha$: 18.40, $\beta$: 0.36 | −1567.9 | Y | 3139.8 | 3139.8 | 3149.3 |
| 3 | nakagami | $\mu$: 4.83,46.13 | −1569.2 | Y | 3142.5 | 3142.5 | 3152.0 |
| 4 | birnbaumsaunders | $\beta$: 6.43, $\gamma$: 0.24 | −1572.8 | Y | 3149.6 | 3149.6 | 3159.1 |
| 5 | inversegaussian | $\mu$: 6.62, $\lambda$:114.96 | −1573.0 | Y | 3150.1 | 3150.1 | 3159.6 |
| 6 | lognormal | $\mu$: 1.86, $\sigma$: 0.24 | −1573.1 | Y | 3150.1 | 3150.2 | 3159.7 |
| 7 | rician | $s$: 6.42, $\sigma$: 1.56 | −1578.1 | Y | 3160.1 | 3160.2 | 3169.6 |
| 8 | normal | $\mu$: 6.62, $\sigma$: 1.53 | −1578.9 | Y | 3161.8 | 3161.9 | 3171.3 |
| 9 | tlocationscale | $\mu$: 6.62, $\sigma$: 1.53, $\nu$: 2594780.26 | −1578.9 | Y | 3163.8 | 3163.9 | 3178.1 |
| 10 | loglogistic | $\mu$: 1.87, $\sigma$: 0.14 | −1584.9 | Y | 3173.8 | 3173.8 | 3183.3 |
| 11 | logistic | $\mu$: 6.56,$\beta$: 0.88 | −1590.1 | Y | 3184.2 | 3184.2 | 3193.7 |
| 12 | uniform | $a$: 2.89, $b$: 11.24 | −1814.4 | N | 3632.8 | 3632.8 | 3642.3 |
| 13 | rayleigh | $b$: 4.8 | −1946.1 | N | 3894.2 | 3894.2 | 3898.9 |
| 14 | gp | −0.98, $\theta$: 11.01 | −2068.1 | N | 4140.2 | 4140.2 | 4149.7 |
| 15 | exponential | $\theta$: 6.62 | −2470.5 | N | 4943.1 | 4943.1 | 4947.8 |

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
