# Peer review of "Bicycle Speed Modelling Considering Cyclist Characteristics, Vehicle Type and Track Attributes"

_wevj, doi:10.3390/wevj12010043_

Round 1

Reviewer 1 Report

Xingchen Yan, Xiaofei Ye, Jun Chen, Tao Wang, Zhen Yang and Hua Bai

Bicycle Speed Modelling Considering Cyclist, Vehicle and Track Heterogeneity

Abstract

In the abstract, it needs to be clarified, which speed the bike refers to? Riding speed or travel speed?

  1. Introduction

It needs to be clarified which speed change the bicycle refers to. It can only be concluded that the driving speed was examined: „we only consider the cycling speed in the free-flow state”.

According to Hungarian measurements, for electric bicycles that assist up to 25 km/h, the ride speed generally did not increase for conventional bicycles. The maximum speed was lower for electric bicycles, which is higher after the cessation of assistance, which can be explained by its mass.

In contrast, on routes where decelerations, accelerations, stops are shared, electric bicycles' average speed exceeds the average speed of human-powered bicycles.

There are also significant differences between electric bicycles. In Europe, electric bicycles are assisted up to a speed of 25 km/h, the electric drive cannot operate on its own, it only helps to drive the bicycle. Vehicles that are assisted at speeds above 25 km/h are already considered to be electric mopeds and are therefore subject to different traffic rules (e.g. they are not allowed to cycle on the road in built-up areas).

  1. Methodology

The method used is adequate, and the description is easy to understand.

The figures are well related to the explanation.

  1. Data Preparation and Description

The application of the method is appropriate. The figure is illustrative.

  1. Clustering results

The presentation of the calculations is understandable, the figures are correct.

  1. Distribution Fitting Results for Speed Clusters

The presentation of the results is understandable. The figures are illustrative.

There is only a reference to the 25 km/h speed limit for electrical assistance, but this is particularly important in Europe.

Conclusion

The speed of the bicycle has been studied based on five factors. However, in practice, uninterrupted traffic is rare. Usually, the actors of the traffic also influence each other.

Therefore, the study results cannot be used directly. However, the method is suitable for further development, taking into account other influencing factors.

I suggest examining how sensitive the disturbances are to the velocities found at free flow.

References

There is a wealth of available literature on the subject, so it isn't easy to select the most important ones. The documents referenced in the publication are in line with the research topic.

Author Response

Dear Professor,

We are truly grateful to your critical comments and thoughtful suggestions. Based on these comments and suggestions, we have made careful modifications on the original manuscript. All changes made to the text are highlighted in yellow. We hope the new manuscript will meet your requirements. Below you will find our point-by-point responses to your comments:

  1. The cycling speed in the study refers to free riding speed which was described in detail in line 83-85. Besides, the related descriptions in the revised manuscript had been updated.
  2. E-bike-related traffic laws and directives of the regions outside China, especially Europe, had been reviewed and referred to support the discussions, please see line 383-386.
  3. The aim of the paper mainly provides a useful base for bicycle-related traffic simulations and theoretical derivations rather than a direct application in practice. This point was clarified in the abstract line 18 of the new version and the arbitrary statement had been eliminated, see line 409-410.
  4. Thanks for your valuable suggestions on the sensitive analysis on cycling speed, we had conducted such a study in another paper, which was described in line 101-103.
  5. Besides, the paper title was revised to reflect the study accurately.

Thank you again for your time and consideration.

Sincerely,

Authors

Reviewer 2 Report

Authors present a paper written in rather good English (however, there are several grammar mistakes). All parts of the paper are easily understood. However, it is not clear what is the general idea of this research in context of elecrtic vehicle control/development etc.

The authors perform cluster analysis of some data (where are these data from, how are they collected? why authors use binarization of real features such as age or lane width?) using the k-means algorithm, and then they solve the problem of identification of the distribution of the features for each of clusters.

Why the authors need clustering for solving this problem? Why the authors use the k-means methods to separate probebility distributions? There exists the EM algorithm and analogous mix probability distribution separation.

It seems like the authors performed this data analysis in accordance with some pre-defined guidelines, and all this research reminds students' lab or course work on cluster analysis.

In Table A2, the result of clustering is definitely incorrect: obviously, there must be 2 clusters, not 3.

The bibliopgraphy is short and does not reflect the state-of-the art in vehicle control nor data processing nor clustering.

The authors twice refer to their previous research results but they give no idea which research they mean and where a reader can find the results.

Overal recommendation: reject or major revision

Author Response

Dear Professor,

We are truly grateful to your critical comments and thoughtful suggestions. Based on these comments and suggestions, we have made careful modifications on the original manuscript. All changes made to the text are highlighted in yellow. We hope the new manuscript will meet your requirements. Below you will find our point-by-point responses to your comments:

  1. Motivation of the study was to provide a useful base for bicycle-related traffic simulations and theoretical derivations which was clarified in the abstract (line 18) of the new version. Research content was aiming at the journal’s scope I-1 “Modelling & Simulation” and the subject was the speed output of bicycles riding within separate lanes rather than electric vehicle control.
  2. To keep the brevity of the paper, the data collection indicated in detail in our previous study (reference 10) was listed shortly in the first version of manuscript. According to your suggestion, some necessary indications and descriptions had been added into the new version, please see line 101-107.
  3. Using binarization of real features was mainly because most of factor variables influencing cycling speed were category variables like gender and bicycle type. The ages of cyclists can only be estimated in the process of data extraction from the operation videos. Thus, we had to divide the ages into five age groups to reduce estimation errors. Speed data under different lane widths was examined by multi-comparison and merged into few width intervals represented in numbers.
  4. The speed data do not present a bimodality or multimodality, so unimodal distributions were fitted to the data. To indicate this, bimodality coefficient calculation was performed in addition, please see line 109-116 and line 331-333.
  5. Thanks for your careful checking, the data of paper had been fully reviewed and some errors in clustering process had been corrected. The results was updated now, please see Table A2, Figure 6. The new results keep the optimal number 3 indicated by Figure 6 and confirmed by another similar study by Xu et al. Please see line 322-324.
  6. As listed above, the subject of the paper was the speed characteristics of bicycles and thus we only collected the references closely related to the subject.
  7. Besides, we had checked and corrected the grammar errors in the manuscript, and the paper title was revised to reflect the study accurately.

Thank you again for your correction. We know that every submission is a valuable learning opportunity, your guidance has greatly improved our paper.

Sincerely,

Authors

Round 2

Reviewer 1 Report

I accept the amended article.

Author Response

Dear Professor,

We are truly grateful to your approval of our new version of manuscript. I wish you a happy work and life!

Thank you again for your time and consideration.

Sincerely,

Authors

Reviewer 2 Report

Authors actually made some minor corrections and improvements.

However, most of the comments for the previous version are still actual.

Comments to the authors' response:

1. Motivation of the study was to provide a useful base for bicycle-related traffic simulations and theoretical derivations which was clarified in the abstract (line 18) of the new version. Research content was aiming at the journal’s scope I-1 “Modelling & Simulation” and the subject was the speed output of bicycles riding within separate lanes rather than electric vehicle control.

This version of the abstract is better.

2. To keep the brevity of the paper, the data collection indicated in detail in our previous study (reference 10) was listed shortly in the first version of manuscript. According to your suggestion, some necessary indications and descriptions had been added into the new version, please see line 101-107.

Nevertheless, some general characteristics of data such as their volume and  dimensionality must be clearly indiceted. Without such characteristics, the necessity of such data processing techniques as clustering are not substatniated.

Why the authors need clustering for solving this problem? Why the authors use the k-means methods to separate probebility distributions? There exists the EM algorithm and analogous mix probability distribution separation.

3. Using binarization of real features was mainly because most of factor variables influencing cycling speed were category variables like gender and bicycle type. The ages of cyclists can only be estimated in the process of data extraction from the operation videos. Thus, we had to divide the ages into five age groups to reduce estimation errors. Speed data under different lane widths was examined by multi-comparison and merged into few width intervals represented in numbers.

If the data contain five age groups (authors can use the median age value for each of age groups or enumerate them 1..5), their direct using must give much more accurate results than their binarization which results in two groups. The same about the speed. It is still not clear why authors use binarization.

6. As listed above, the subject of the paper was the speed characteristics of bicycles and thus we only collected the references closely related to the subject.

The bibliography was slightly improved but it is still insufficient.

Author Response

Dear Professor,

We are truly grateful to your critical comments and thoughtful suggestions. Based on these comments and suggestions, we have made careful modifications on the original manuscript. All changes made to the text are highlighted in yellow. We hope the new manuscript will meet your requirements. Below you will find our point-by-point responses to your comments:

  1. Basic characteristics of field survey samples was added in Table 1, which describe the data volume and dimensionality, please see line 160.
  2. To clearly explain the reason of using clustering and K-means clustering other than EM algorithm and analogous mix probability distribution separation, a subpart named “2.1.1 study logic” was presented. The first paragraph (line 100-115) indicated why clustering was needed. The rest two paragraphs reviewed clustering techniques and especially the third focused on the reason that EM algorithm and other mix probability distribution separation were not suitable for the study (please see line 134-139).
  3. Using binarization of real features was a trade-off due to the limitations of the data extraction technology we mastered at that time of investigating. We initially wanted to get the exact ages of the cyclists. It was the limitations of the study. We will solve this problem in future research using AI technologies like object and character recognition. The limitations above were described in the final (please see line 460-463).

Sincerely,

Authors

Round 3

Reviewer 1 Report

I accept the amended article.

Author Response

Dear professor,

Thank you for your recognition, and good luck with your work!

Reviewer 2 Report

This version of the paper is much better because of a new section. However, the style of this new section MUST be improved.

The Bibliography was significantly improved, however, a further improvement is possible. For example, the authors describe the EM algorithm as a clustering tool only. However, the EM algorithm is a method for probability distribution separation. It is able to separate distribution and simultaneously search for their unknown parameters which is probably much more important in the context of this paper.

Author Response

Dear Professor,

We are truly grateful to your critical comments and thoughtful suggestions. Based on these new comments and suggestions, we have made careful modifications on the original manuscript. All changes made to the text are highlighted in yellow. We hope the latest manuscript will meet your requirements. Below you will find our point-by-point responses to your comments:

  1. More applications of EM algorithm were added into the latest manuscript and its good performance in parameter estimation was highlighted specially. Please see line 128-133.
  2. Besides, grammar errors had been corrected in the manuscript.

Sincerely,

Authors
